# Cranial MRI in Childhood Acute Leukemia during Treatment and Follow-Up Including the Impact of Intrathecal MTX—A Single-Center Study and Review of the Literature

**DOI:** 10.3390/cancers14194688

**Published:** 2022-09-26

**Authors:** Marvin Mergen, Sascha Scheid, Hannah Schubmehl, Martin Backens, Wolfgang Reith, Norbert Graf

**Affiliations:** 1Department of Pediatric Oncology and Hematology, Saarland University, 66421 Homburg, Germany; 2Department of Diagnostic and Interventional Neuroradiology, Saarland University, 66421 Homburg, Germany

**Keywords:** cerebral MRI, childhood acute leukemias, ventricular width, morphologic alterations, intrathecal MTX

## Abstract

**Simple Summary:**

Acute leukemias in children have an excellent outcome. Therefore, reasons for long-term sequelae need to be discovered to avoid them. Our retrospective analysis in addition to a literature review focuses on neurotoxicity by investigating cerebral morphologic changes diagnosed in cranial MRIs in children with acute leukemias. Such changes occur in more than 50% of children. They were detected especially in patients below 6 years of age, in ALL and in patients receiving more than 12 intrathecal MTX applications. The ventricular width, as one of the cerebral abnormalities, has the potential to normalize again. It is important to prospectively investigate and correlate these changes with the neurocognitive outcome in children with acute leukemias to evaluate the impact of intrathecal MTX in this context.

**Abstract:**

Due to high survival rates, long-term sequelae, especially neurotoxicity, need to be considered in childhood acute leukemias. In this retrospective analysis of morphologic changes of the brain in children treated for acute leukemias, we included 94 patients (77 ALL, 17 AML; 51 male, 43 female; median age: 5 years) from a single center. We analyzed 170 cranial MRI scans (T2, FLAIR axial) for morphologic alterations of the brain and variations of the ventricular width (GDAH). In addition, the corresponding literature was reviewed. More than 50% of all patients showed cerebral pathomorphologies (CP). They were seen more often in children with ALL (55.8%), ≤ 6 years of age (60.8%), in relapse (58.8%) or after CNS irradiation (75.0%) and included white matter changes, brain atrophy, sinus vein thrombosis and ischemic events. GDAH significantly enlarged mainly in children up to 6 years, with relapse, high-risk leukemias or ALL patients. However, GDAH can normalize again. The number of intrathecal Methotrexate applications (≤12 vs. >12) showed no correlation to morphologic alterations besides a significant increase in GDAH (−0.3 vs. 0.9 mm) between the first and last follow-up MRI in ALL patients receiving >12 ith. MTX applications. The role of ith. MTX on CP needs to be further investigated and correlated to the neurocognitive outcome of children with acute leukemias.

## 1. Introduction

Acute lymphoblastic leukemia (ALL) is the most common childhood cancer [1]. Over the last decades, the outcome of children with ALL and AML has improved significantly, mainly based on multicenter randomized trials. Today, 90% of children with ALL and 70% of children with AML can be cured [1]. Besides the effectiveness of the current treatments, short- and long-term sequelae need to be taken seriously. Deterioration of neurocognitive function is one of the most relevant late effects [2,3,4,5,6,7,8,9,10,11,12,13]. To analyze the reason for declined neurocognitive abilities, imaging studies of the central nervous system (CNS) are helpful to find correlated morphological changes. In contrast to adults [14], this is especially important in children as ALL and AML occur at a median age still in a vulnerable phase of brain development. At that time, they receive potential neurotoxic drugs, and some of them also receive cranial irradiation to treat or to prevent CNS leukemia [15,16,17,18,19,20,21,22,23,24,25,26,27,28]. Interestingly, large-scale clinical trials analyzing the effects of treatment on cranial MRI (cMRI) in these children are rare.

The aim of this study is to investigate cerebral morphologic changes by cMRI in children with acute leukemias and to identify possible risk factors in comparison with a review of literature.

## 2. Materials and Methods

We analyzed cMRI scans of children with acute leukemias retrospectively. All children were treated at the Department of Pediatric Oncology and Hematology at Saarland University Medical Center in Homburg, Germany, between 2007 and 2017. Patients underwent cMRI at diagnosis and later only when neurologic symptoms occurred during treatment or afterwards. Of the 94 included patients, 77 (81.9%) were diagnosed with ALL and 17 (18.1%) with AML. Patients were classified into high and intermediate risk according to the criteria defined by the trial protocols in which patients were enrolled [29,30,31,32]. Altogether 17 (18.1%) patients suffered from recurrence during follow-up with a median age at relapse of 7.7 years. Further details of patients’ characteristics are given in Appendix A.

All patients received their first MRI up to 30 days after diagnosis (induction period P1). Those without neurological symptoms at any time (47 patients; 50%) received only one cMRI during P1. In 29 patients (30.9%), two; in 7 (7.4%), three; and in 11, (11.7%) four cMRIs were performed. Overall, we analyzed 170 cMRI scans (146 in ALL, 24 in AML). Neurological symptoms requiring a consecutive MRI were seizures, impaired consciousness, sleepiness, cranial nerve and other palsies, motor weakness, behavior changes and other psychomental abnormalities.

Four follow-up cMRI scans were already performed in 4 patients during the first 30 days (induction period (P1)) after diagnosis, 44 during the period between 30 days and 1 year in 36 patients (reinduction and beginning of maintenance treatment (P2)), and 40 in 31 patients after 1 year (maintenance treatment and follow-up period (P3)). For specific analysis of the ventricular width, we also defined timepoint 1 (t1) as the first cMRI of a patient, timepoint 2 (t2) as the last cMRI in P2, and timepoint 3 (t3) as the last cMRI in P3.

All patients were treated according to BFM protocols for ALL (AIEOP-BFM ALL 2009 [29], ALL-BFM 2000 [30,31]), and AML (AML-BFM 2004 [32]). These trials contain vincristine, intrathecal and intravenous methotrexate (MTX), and intrathecal cytarabine, which are potentially neurotoxic. Only 8 patients received cranial irradiation, of whom 7 showed CNS involvement. One child with T-ALL received prophylactic cranial irradiation. The number of intrathecal methotrexate (ith. MTX) applications was counted for each patient. The dose of ith. MTX was 12 mg for children above 3 years, 10 mg between 2 and 3, 8 mg between 1 and 2, and 6 mg under 1 year of age. In 13 patients with ALL relapse or AML, ith. MTX was combined with intrathecal ARA-C and prednisone.

For subgroup analysis, we retrieved clinical data from the children’s medical records. The cMRI scans were accessed through the hospital’s PACS system. Details of the cMRI modalities are given in Appendix A.

GDAH was investigated in all patients at diagnosis and in addition in those patients with consecutive MRI scans during follow-up. Brain atrophy (a visible increase in the outer cerebral fluid space) was graded by an experienced neuroradiologist (WR) as described by Bermel et al. [33] and previously used by Iuvone et al. [34]. White matter changes (WMC) were graded from 0–3 (0: no WMC, 1: punctate foci, 2: beginning confluence of foci, 3: large confluent areas), according to Fazekas’ classification [35].

In addition, for reviewing CP in children with leukemias, we performed a literature search via Pubmed. The strategy and study selection process are depicted in Appendix A [36].

Intra- and interobserver variability of GDAH was tested by measuring GDAH by two observers (MM, WR) independently in 10 different randomly selected cMRIs including 3 readings by each observer. The mean interobserver variability was 0.37 mm with a mean variance of 0.23 mm and standard deviation of variance of 0.21 mm for observer 1 compared to 0.07 mm and 0.09 mm for observer 2.

Data management and analysis was performed using SPSS, version 25.0 for Mac (IBM SPSS Statistics 27, Armonk, NY, USA) after anonymization of the data, thus adhering to the European General Data Protection Regulation (GDPR).

To test statistical significance, which was defined as *p* < 0.05 (α = 0.05), we used chi square tests for comparisons of morphological changes between subgroups. T-tests for connected and disconnected data were used for analysis of GDAH.

Ethical approval was given by the Institutional Ethics Committee of the ‘Ärztekammer des Saarlandes’ (ID-numbers: 51/06, 15/08, 53/13, 58/13, 101/11) for the involved clinical trials and for accompanying basic and clinical research. Informed consent was obtained from all parents or legal guardians and from patients older than 14 years.

## 3. Results

We looked for cerebral pathomorphologies (CP), such as white matter changes (Figure 1a–d), brain atrophy (Figure 1e), sinus vein thrombosis (Figure 1f), and ischemia (Figure 1g). Ventricular width (GDAH) was measured as the greatest distance between the outside lateral walls of the anterior ventricular horns (Figure 1h). This method is one of the techniques used in monitoring ventricular enlargement of patients with multiple sclerosis [37]. The results are divided into two parts. The first part deals with cerebral pathomorphologies (CP) including GDAH and the second part analyzes the influence of ith. MTX on these CP. Detailed results regarding cerebral CP can be found in Appendix A.

### 3.1. Cerebral Pathomorphologies

#### 3.1.1. General Cerebral Pathomorphologies (CP)

During the whole follow-up time, we detected CP in 48 of 94 patients (51.1%) in 98/170 MRIs (42.1%). In period 1, CP was already found in 46.3% of all patients. This number increased in period 2 to 72.3% and dropped again to 54.8% in period 3. Patients with ALL showed significantly more CP (55.8%) than those with AML (23.5%) (*p* = 0.012). The same was true for patients up to 6 years of age at first diagnosis with 60.8%, compared to 39.5% in older ones (*p* = 0.040). After irradiation, CP was found in 75% versus 48.8% in non-irradiated patients.

A total of 38 patients (40.4%) still showed CP in their last MRI during follow-up. On the other hand, in 20.8% of patients these changes resolved over time, even in two patients belonging to the high-risk group.

#### 3.1.2. White Matter Changes (WMC)

White matter changes (WMC) were detected in in 16/94 patients (17.0%). The maximum intensity was grade 1 in 25.0%, grade 2 in 56.3%, and grade 3 in 18.7% of patients. In 12.2% of patients, they were already found during period 1 and even in two during the first week after diagnosis. If WMC occurred, they never resolved but intensified over time.

WMC were more often found in ALL (19.5%) than AML patients (5.9%). Interestingly, one third of patients with WMC suffered from CNS leukemia. Patients receiving cranial irradiation showed a tendency of more white-matter changes than non-irradiated ones (25% vs. 16.3%).

Grade 1 WMC were seen in 4.9% of patients during period 1, whereas this grade was not seen at all in period 2 or 3. While 7.3% of patients showed grade 2 in period 1, this number decreased to 5.6% in period 2 and increased again to 12.9% in period 3. The number of grade 3 WMC grew over time from 0% in period 1 to 2.8% in period 2 and 6.5% in period 3.

Seven patients had isolated white-matter changes and nine showed other pathologies as well (7× brain atrophy, 1x ischemia, 1x Posterior Reversible Encephalopathy Syndrome (PRES)). In two patients, WMC occurred before BA was diagnosed. Five of seven patients with WMC and BA suffered from bone marrow relapse.

#### 3.1.3. Sinus Vein Thrombosis (ST)

Sinus vein thrombosis (ST) was detected in five (four ALL, one AML) patients (5.3%). Most sinus vein thromboses were seen in period 2 (13.9% of all patients with cMRIs during this period). Three children suffered from ST during period 1, two of them even in the first week after diagnosis. In four patients, ST was diagnosed two months after initial diagnosis. In addition, 4/5 children suffered from ischemic events: 2 years, 6 and 2 months, and 5 weeks after ST. During follow-up ST resolved completely in all patients before the end of leukemia treatment.

#### 3.1.4. Brain Atrophy (BA)

We found brain atrophy (BA) in 37.2% of patients (33 ALL, 2 AML). Out of these 35 patients, BA occurred already in 18 during the first four weeks and in 6 even in the first week after diagnosis. In addition, 15 patients developed BA in period 2. In only 2 patients, BA occurred later during treatment. A significant difference was found when comparing ALL with AML patients (42.9% in ALL vs. 11.8% in AML; *p* = 0.016). There was no significant difference in prevalence of BA related to gender. Patients up to 6 years of age (50.1%) showed a significantly higher rate of BA than older patients (20.9%; *p* = 0.003). In 25 cases, BA was the only detected CP.

#### 3.1.5. Cerebral Ischemia

Ischemia in this cohort of patients is mainly caused by venous infarcts and is found in 7 patients (7.3%), of whom 4 had sinus vein thrombosis before. Most ischemic events occurred within the first 6 months after diagnosis (71.4%). All ischemic events were reversible in cMRI if they were independent of thrombosis. Nevertheless, of these patients, one showed WMC and one BA.

#### 3.1.6. Ventricular Width (GDAH)

Detailed results of GDAH can be found in Figure 2 and Figure 3. The mean GDAH was associated with an overall significant larger ventricular width in males compared to females (+5.7%) (*p* = 0.006) during the first year after diagnosis. No significant differences could be seen comparing age groups (≤ and >6 years) in general. GDAH increased by 5.7% from P1 to P2 (*p* = 0.017) and 3% from P1 to P3 (*p* = 0.028) in patients ≤ 6 years of age. In addition, patients with CP also developed significantly larger GDAH during the first year (+4.8%; *p* = 0.004) and follow up (+7%; *p* = 0.009). Patients without visible BA showed a significant increase of 4.2% from P1 to P2 (*p* = 0.011) and 2.7% from P1 to P3 (*p* = 0.031). Children with relapse had an overall 5.6% larger mean GDAH during follow-up than non-relapsed patients with an increase in GDAH of 3.5% from P1 to P3 (*p* = 0.017). Additionally, high-risk patients showed an individual significant 4.1% increase in GDAH during their treatment and follow-up (*p* = 0.018). Ventricular width could also decrease on an individual level again, as shown in non-relapsed patients from P2 to P3 (−3%).

### 3.2. Intrathecal Methotrexate (ith. MTX) in Patients with Acute Leukemia

Altogether, 1393 ith. MTXs applications were given to 94 patients (Table 1).

The highest number of ith. MTX was applied to patients with ALL relapse and CNS involvement. Significantly more ith. MTX was given to patients with ALL compared to AML. In case of ALL, a significantly different distribution of the number of ith. MTX applications (≤12 vs. >12) was found for risk group, relapse, hematopoietic stem cell transplantation (SCT), CNS involvement and gender. However, more than 12 ith. MTX applications did not correlate with BA, WMC, ST and ischemia (Table 2).

In consecutive cMRIs, the mean and median difference between GDAH at timepoint 3 (t3) and timepoint 1 (t1) being calculated for each patient showed a significant increase in size after more than 12 ith. MTX applications (Table 3). This was not seen during the first time period (≤ 30 days) or between t1 and t2 independent of the number of ith. MTX applications. It is of interest that during the time period > 30 days with less than 12 MTX applications, GDAH can even become smaller (mean −0.3 mm) again compared to the initial GDAH size (t3-t1).

## 4. Discussion

The risk of developing neurocognitive impairments during long-term follow-up in children with acute leukemias is well known, and not only in those children receiving cranial irradiation [8]. The main cause is treatment with potential neurotoxic drugs during the vulnerable phase of brain maturation [38,39,40]. The literature can be divided into two groups, firstly about morphologic changes of the brain in children treated for acute leukemias alone [15,16,17,18,19,20,21,22,23,24,25,26,27,28,41,42] and secondly about the association between cMRI alterations and neurocognitive sequelae [2,3,4,5,6,7,8,9,10,11,12,13,22,34,43,44,45,46].

In this study, we analyzed only morphological changes including GDAH in cMRI in 94 children with acute leukemias without a correlation to neurocognitive outcome. To the best of our knowledge, this is one of the largest single-center cohorts described yet.

As follow-up cMRI scans were only performed if neurologic symptoms occurred, we found that in half of the patients no further MRI scans than the one at diagnosis was needed. Nevertheless, in half of our cohort, neurological symptoms of any kind resulted in further cMRI scans. In these patients, we could demonstrate that the prevalence of CP in general is significantly higher in patients with ALL and below the age of 6 years, whereas in females, in the case of cranial irradiation or relapse, patients show only a tendency of more CP. GDAH increases mainly in boys and children up to 6 years of age, in those with ALL, high-risk leukemia, recurrence and if other pathomorphologies are detected in cMRI. An isolated impact on CP cannot be answered accurately for patients with CNS involvement and cranial irradiation as their numbers are too small. However, the increase in GDAH in children with ALL compared to AML is explainable by the higher number of ith. MTX applications in ALL and the potential neurotoxic dexamethasone [28] that is only given in ALL. Unfortunately, the number of patients was too small to compare the influence of the type of corticosteroid (dexamethasone versus prednisone, ALL-BFM 2000) or the prolongation of asparaginase (AIEOP-BFM ALL 2009) in randomized ALL trial patients. No comparison of GDAH between ALL and AML is found in the literature. The limited number of only 17 patients with AML in our cohort allows no further definitive conclusions compared to ALL. To find causes for specific rare CP, such as sinus vein thrombosis or ischemia, our results are limited as well.

Comparing our results with the literature review, we agree with Svärd, who stated: “Results in neuroimaging studies on ALL survivors are highly variable, largely because of methodological variations and limitations such as small samples, different treatment protocols, and different follow-up times” [13]. Nevertheless, our main findings and those of the literature, summarized in Table 4 and Table 5 and Appendix A, allow us to draw the following conclusions dealing with CP in cMRI in children with acute leukemias.

Like us, Porto et al. [19] found CP in 15 of 21 ALL/AML patients in the first 3 months. The rate of long-term findings was high in our analysis and in that of Hertzberg et al., 1997 [7], who detected most CP 5–8 years after treatment. Besides gender in our study, we and others did not find any other significant differences in the prevalence of CP [41]. Age as a risk factor is not a common finding in the literature [2,7,41]. Like us, Rijmenams et al. [42] identified young age as a risk factor for leukoencephalopathy.

Comparing ALL and AML, Porto et al. [19] detected more CP in AML (28%) than in ALL patients (14%). Their small number (7 AML- and 14 ALL-patients) cannot be regarded as representative and is in contrast to our findings in a larger cohort with a significantly higher prevalence of CP in ALL than in AML patients. Differences according to risk groups are varying, and results from Anastasopoulou et al. [2] and Badr et al. [41] show a higher incidence in the high-risk group.

Our results showed white-matter changes less often than Rijmenams et al. [42], Duffner et al. [5], and Iuvone et al. [34], who detected this pathology in 41–67%. Nevertheless, some of these cohorts more often received cranial radiotherapy. Pääkkö et al. [24] found WMC only in irradiated patients.

WMC never resolved but partly intensified during treatment of childhood leukemia. This was especially true for severe forms. With our median follow-up time of 7.7 years, this finding is in line with Duffner et al. [5] seeing remaining changes after this time period, and Dellani et al. [20] even after 16–28 years.

In our study, all sinus vein thromboses dissolved after one year, matching the results from the literature [23]. As sinus vein thrombosis can cause congestion bleeding in the brain, a connection between this pathology and ischemic events occurred in our cohort in 4/7 patients.

Brain atrophy was especially seen in our cohort below 6 years of age, confirming that the brain at a young age is more vulnerable to neurotoxic effects. A reduction in brain volume in ALL survivors caused by leukemia treatment is described by Van der Plas et al. [43,44], Morioka et al. [15], and Dellani et al. [20]. Nevertheless, results from the literature are inconsistent. In contrast, no significant difference in cerebral volume could be found between survivors of ALL and controls after 5–10 years by Philips et al. [28]. However, they described smaller hippocampi, cerebelli and thinner cortices in different brain regions, matching the results of Zajac-Spychala et al. [4]. These observations correlated with the dose of dexamethasone, and female patients were more affected than male patients. Comparisons between ALL and AML patients and between younger and older children were not found in the reviewed literature.

There are only a few publications that address ventricular width in children with leukemia. Besides one [28], all others are published before 2000 and focus on irradiated patients or those with CNS relapse, rare subgroups today (Table 5). A study of Kretzschmar et al. [27] found increased ventricles one year after CNS therapy in 8/21 patients (mainly patients with CNS relapse). They could also show that ventricular size can decrease again after 2–8 years. Pääkkö et al. [24] found larger ventricles in 37% of their 27 ALL patients. However, only two patients did not receive cranial radiotherapy, which is why the comparison between irradiated and not-irradiated patients is difficult to interpret. Additionally, ventricular size at diagnosis is not given. Two years later, Pääkkö et al. [25] detected a slight-to-medium enlargement of ventricles and increased sulci associated with ALL therapy. Similar results were found in a study by Hertzberg et al. [7], who saw more ventricular enlargement in irradiated compared to non-irradiated patients (36.8% vs. 24.4%). Due to the reduction in preventive cranial radiotherapy in modern treatment protocols, only 8 patients in our cohort were irradiated, and this number is too small to analyze this question.

An increase in ventricular width can be reversible in some patients related to a normalization of fluid balance after treatment with steroids but can also be irreversible due to a loss of white matter.

One reason for neurotoxicity is ith. methotrexate as described by several authors [22,25,47,48]. We could show that more than 12 ith. MTX applications did increase GDAH in patients with ALL over time. This is a finding that is not documented by others. On the other hand, more than 12 ith. MTX applications are also given to high risk, relapsed, and CNS-positive ALL patients, which is why the influence of a more aggressive treatment cannot be ruled out to be responsible for larger ventricles in combination with ith. MTX. Gandy et al. [8] reviewed 23 articles dealing with brain imaging of ALL survivors and stated that even i.v. chemotherapy can have neurotoxic sequelae by damaging the blood–brain barrier (BBB), inducing the apoptosis of brain cells, directly harming DNA, increasing oxidative stress, shortening the telomere length, and impairing neurogenesis.

As all patients in our cohort were investigated in a single center and analysis was carried out by two independent observers (M.M., W.R.), our results can be regarded as reliable and valid. Nonetheless, larger cohorts with standardized MRI procedures are needed to understand cerebral pathomorphological changes and the relevance of GDAH in rare subgroups. As our observations are retrospective, selection bias is possible. In our study, neuroimaging was performed at diagnosis and during follow-up only in patients if neurological symptoms occurred. To reduce such a selection bias, prospective standardized cMRI-follow-up protocols in children with leukemia are needed to detect cMRI abnormalities also in those patients without neurological symptoms.

As mentioned, morphologic changes can come along with neurocognitive impairment in children with leukemia. The association between morphologic changes to the brain and neurocognitive outcome was not the intention of this retrospective analysis. Nevertheless, findings from the literature are also summarized in the Appendix A. Further studies are needed to specifically address this topic. 

## 5. Conclusions

In summary treatment for childhood leukemia can cause cerebral pathomorphologic changes, an increase in ventricular width, smaller hippocampi and cerebelli and thinner cortices in different brain regions. Risk factors related to leukemia itself (ALL, relapse, CNS involvement), to the characteristics of patients (young age), and to treatment (ith. MTX, dexamethasone, cranial irradiation) could be identified. The number of ith. MTX applications in particular plays a role in the enlargement of ventricular width. Based on this study and the available literature, prospective trials are needed to confirm these results. In addition, correlations between CP detected in cMRI and neurocognitive changes after leukemia treatment should be included in such trials to try to prevent long-term CNS sequelae in future without jeopardizing the excellent survival.

## Figures and Tables

**Figure 1 cancers-14-04688-f001:**
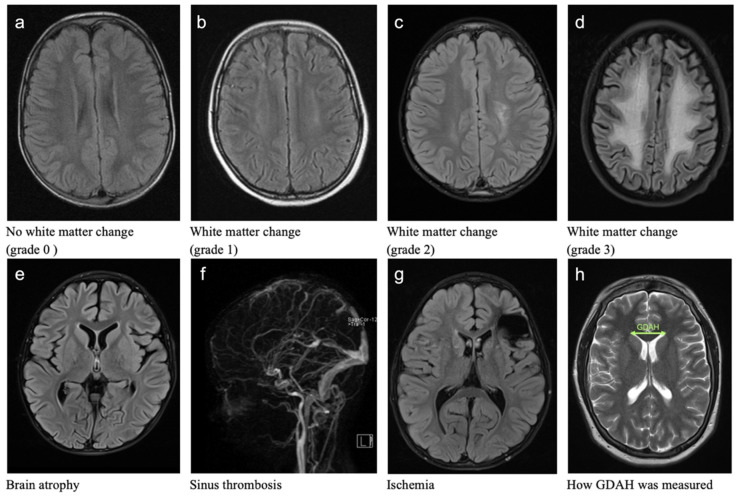
CMRI axial: Examples of different pathomorphologies and how GDAH was measured. (**a**): No white matter change (grade 0); FLAIR; (**b**): white-matter change (grade 1); FLAIR; (**c**): white-matter change (grade 2); FLAIR; (**d**): white-matter change (grade 3); FLAIR; (**e**): brain atrophy; FLAIR; (**f**): sinus vein thrombosis; contrast enhanced angiography, sagittal; (**g**): ischemia; FLAIR; (**h**): how GDAH was measured; T2.

**Figure 2 cancers-14-04688-f002:**
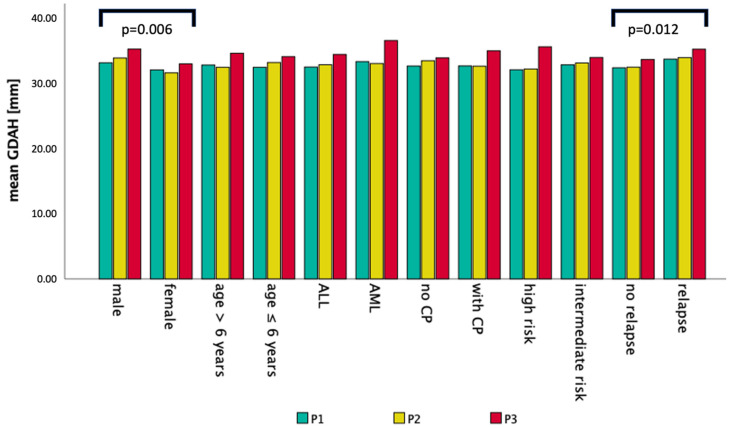
Mean ventricular width (GDAH) of all MRIs is displayed for different time periods (P1, P2, P3) and categories (CP: cerebral pathomorphologies).

**Figure 3 cancers-14-04688-f003:**
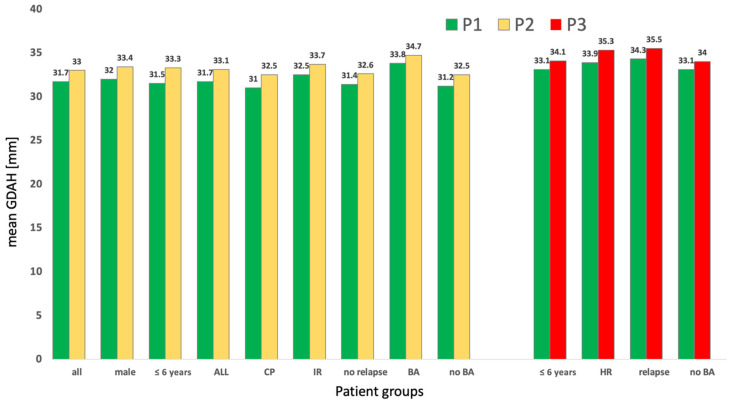
The change in mean ventricular width (GDAH) in individual patients between different time periods (P1, P2 and P3) is displayed. Only significant changes (*p* < 0.05) are shown (CP: cerebral pathomorphologies; IR: intermediate risk; HR: high risk; BA: brain atrophy).

**Table 1 cancers-14-04688-t001:** Number of intrathecal (ith.) MTX applications in ALL and AML.

n MTX/Patient		ALL				AML				All			
		Patients		Sum MTX		Patients		Sum MTX		Patients		Sum MTX	
		n	%	n	%	n	%	n	%	n	%	n	%
≤12	7					12	70.6	84	43.7	12	12.8	84	6.0
	12	53	68.8	636	53.0	1	5.9	12	6.3	54	57.4	648	46.5
>12	14					1	5.9	14	7.3	1	1.0	14	1.0
	15	10	13.0	150	12.5					10	10.6	150	10.8
	16	1	1.3	16	1.3					1	1.1	16	1.1
	20					1	5.9	20	10.4	1	1.1	20	1.4
	28	4	5.2	112	9.3					4	4.3	112	8.1
	31	6	7.8	186	15.5	2	11.8	62	32.3	8	8.5	248	17.8
	32	2	2.6	64	5.3					2	2.1	64	4.6
	37	1	1.3	37	3.1					1	1.1	37	2.7
sum		77	100	1201	100	17	100	192	100	94	100	1393	100
Mean MTX/patient				15.6	*p* = 0.000			11.3				14.8	

**Table 2 cancers-14-04688-t002:** Number of ith. MTX applications (≤12 vs. >12) in patients with ALL and their correlation to risk factors and morphological changes in cMRI. (SCT: hematopoietic stem cell transplantation).

Subgroup	Number of ith. MTX Applications		Chi-Quadrat Test
	≤12 [n]	>12 [n]	
**Gender**			
Male/Female	33/20	9/15	0.038
**Risk Group**			
IR/HR	52/1	7/17	0.000
**Relapse**			
No/Yes	53/0	11/13	0.000
**SCT**			
No/Yes	53/0	9/15	0.000
**CNS positive**			
No/Yes	52/1	19/5	0.010
**CNS irradiation**			
No/Yes	51/2	20/4	0.072
**Brain atrophy**			
No/Yes	29/24	15/9	n.s
**White matter changes**			
No/Yes	44/9	18/6	n.s
**Sinus vein thrombosis**			
No/Yes	52/1	22/2	n.s
**Ischemia**			
No/Yes	51/2	21/3	n.s

**Table 3 cancers-14-04688-t003:** Ventricular width (GDAH) at different timepoints (t1, t2 and t3) and their differences depending on the number of ith. MTX applications. Highlighted numbers are significantly different (*p* < 0.05).

Ventricular Width (GDAH)	Time Period ≤ 30 Days		Time Period > 30 Days	
	Number of ith. MTX Applications		Number of ith. MTX Applications	
	≤12 [n]	>12 [n]	≤12 [n]	>12 [n]
Mean at timepoint 1 (mm)	32.9	33.4	33.6	34.6
Mean at timepoint 2 (mm)	33.7	34.3	34.4	36.3
Mean at timepoint 3 (mm)	32.0	35.9	33.3	35.5
Median at timepoint 1 (mm)	32.6	32.6	33.1	34.8
Median at timepoint 2 (mm)	35.3	33.9	35.7	36.3
Median at timepoint 3 (mm)	32.8	35.1	32.9	35.6
Mean difference (t3-t1) (mm)	1.8	1.0	**−0.3**	**0.9**
Mean difference (t2-t1) (mm)	−0.8	−1.4	1.7	0.8
Median difference (t3-t1) (mm)	1.1	0.4	**0.0**	**0.8**
Median difference (t2-t1) (mm)	0.0	0.0	1.4	1.1

**Table 4 cancers-14-04688-t004:** Summary of pathomorphologies without ventricle width (GDAH) based on literature. (Age and Follow-up time: (d: day; m: months; y: years); BA: brain atrophy; BBB: blood brain barrier; CNS: central nervous system; DTI: diffusion tensor imaging: DKI: diffusional kurtosis imaging; fMRI: functional MRI; HR: high risk; ith.: intrathecal; n.d.: not done; MRS: magnetic resonance spectroscopy; NT: neurotoxicity; PREST: posterior reversible leukoencephalopathy syndrome; SIADHS: syndrome of inappropriate antidiuretic hormone secretion; ST: sinus vein thrombosis; TIT: intrathecal triple drug (MTX, cytarabine, prednisone); WMC: white matter changes).

Summary of Pathomorphologies in ALL									
	Patients	Age	Follow-Up Time	Specific Imaging Findings	Neurological Findings and Symptoms	Percentage of Pathologic MRIs	Explanation	Recommendations	References
WMC	4483	0 y–19 y	0.1 y–65 y	leukoencephalopathy (frontal and temporal), hyperdense regions, calcifications, meningeal enhancement, grey matter changes, smaller hippocampus and impaired microstructural white matter integrity in frontal brain regions, impaired white matter integrity, altered functional connectivity, microstructural damage in white matter, fornix, uncinate fasciculus, and ventral cingulum MRS, DTI, DKI are more sensitive than cranial MRI	headache, seizures, change in mental status neurocognitive deficits (vocabulary, memory, learning capacity, spatial ability, executive functions, and attention), lower IQ-performance, speech disorders, disorder in fine motor skills, coordination, widespread reductions in brain activation during cognitive tasks, poorer memory and fine-motor functioning outcome, long-term neurobehavioral problems, no significant relationships between MRI outcome and test scores, school placement, or education level, no correlation to neurocognitive impairment	4–100% In 78.8% persistence	leukemia, relapse, low age, treatment, cranial irradiation, chemotherapy, i.v. MTX, ith. MTX, dexamethasone, infections	restrictive use of MTX, earlier leucovorin rescue exploiting multiple MRI (fMRI, DTI, DKI) techniques, monitoring of intracerebral changes throughout therapy and during long-term follow-up to evaluate integrity of white matter regular psychological and clinical follow up longitudinal studies combining neuroimaging and neurocognitive outcome cognitive and behavioral interventions, rehabilitation of children with treatment-associated cognitive impairment is essential studies to genetic polymorphism for risk factors	[5,6,8,10,11,12,15,17,18,19,22,23,34,41,42,45,46]
**ST**	112	3 y–16 y	9 d–6 y	superior sagittal sinus, sigmoid sinus	headache, seizures, hemiparesis, change in mental status	2–14%	leukemia, relapse, treatment, infection	early diagnosis studies to genetic polymorphism for risk factors	[19,23]
**BA**	495	0.3 y–21.7 y	0.1 y–28 y	decreased hippocampal, thalamus, temporal, occipital lobe, nucleus caudatus and cerebelli volume, grey and white matter atrophy	cognitive impairment, lower IQ, poorer verbal abilities, disorder in fine motor skills and coordination, more poorly in working memory and response inhibition, no significant correlation to imaging (2 studies) correlations between working memory and volume of amygdale, thalamus, striatum, and corpus callosum (1 study)	4–100%	leukemia, ith. MTX, (high dose) chemotherapy, dexamethasone, cranial irradiation, females	lower doses of dexamethasone for younger females, NMDA receptor antagonist, avoid irradiation regular psychological and clinical follow up large scale studies are needed to establish time-course of changes for understanding	[3,4,6,18,20,28,41,43,44]
**Ischemia**	25	6.9 y ± 3.0 y	6 y	old infarct and hemorrhage		4%	cranial irradiation, HR	prospective studies	[41]
**Other**	2787	1 m–17.9 y	10 d–37 y	PRES, stroke, hemorrhage meningioma, osteoma CNS lymphoma, inflammation, infections, no altered fMRI activity	seizures, visual disturbances, conscious disturbances, dizziness, headache, fever, ataxia, flaccid paralysis, altered mental status, neurocognitive impairment, longer response times and reduced accuracy performance during cognitive interference processing, no effect on IQ and cognitive development, no correlation to imaging	1.5–28.6%	older age, T-cell ALL, relapse, CNS involvement, HR, induction chemotherapy, TiT, cranial irradiation, infection	avoid irradiation early identification to prevent late effects further prospective studies studies to genetic polymorphism for risk factors	[2,9,13,16,19,21,23,41]
**Summary of pathomorphologies in AML**									
	**Patients**	**Age**	**Follow-up time**	**Specific imaging findings**	**Neurological findings and symptoms**	**Percentage of pathologic MRIs**	**Explanation**	**Recommendations**	**References**
**WMC**	**No reports**								
**ST**	5	3 y–16 y	under treatment	sigmoid sinus	Seizure, limb weakness	1/5	leukemia, treatment	early diagnosis	[23]
**BA**	**No reports**								
**Ischemia**	5	0.6 y–13 y	under treatment	disseminated tiny lesions in thalamus and cerebral white matter	encephalopathy	1/5	leukemia, treatment, infection	early diagnosis	[23]
**Other**	55	0.6 y–16 y	0 d–5 y	disseminated microinfarcts, vasculopathy, hemorrhage, infections, aspergillosis, late: chordoma and other tumors	headache, ataxia, dizziness, altered mental status, hemiplegia, seizures, fever, sepsis	24–28.6%	treatment, leukemia, CNS involvement, relapse, infection	early diagnosis studies to genetic polymorphism for risk factors	[19,23]

**Table 5 cancers-14-04688-t005:** Summary of studies dealing with ventricular width in patients with ALL. (CNS: central nervous system; ith.: intrathecal; WMC: white matter changes).

Patients (n)	Age at Diagnosis (Years)	ith. MTX Applications (n)	Increase in Ventricular Width (No. or % of Patients)	Other Findings	Follow-Up Time	Explanation	Recommendation	Reference
118	Mean: 5.8 0.3–16.1	6–8	37% with irradiation 24% without irradiation	Any MRI abnormality: 61/118 patients (51.7%)	7 years	Cranial irradiation, ith. MTX	Prospective studies	[7]
27	Mean: 3.6 0.3–14	unknown	10/11 (irradiated patients)	In irradiated patients: -4 patients.: WMC -2 patients: meningioma -8 patients: low/heterogeneous intensity foci	2–20 years	Cranial irradiation	No specific recommendation	[24]
18	Mean: 5.5 2.4–14.4	Yes, but unknown n of applications	13 (transient)	2 patients (CNS negative) with WMC	1–29 months	Steroid treatment ith. MTX	No follow-up with MRI is indicated	[25]
28	3.9–14.4	6, but also 24 Gy cranial irradiation	2 patients with severe cerebral atrophy had enlarged ventricles	12 patients: slight atrophic changes 9 patients: severe cerebral atrophy	1–10 years	Disease severity and treatment	Restrictive cranial irradiation	[26]
60	1–14	unknown	10 increased at diagnosis 8/21 1 year after CNS treatment	No cranial pathomorphologies	up to 8 years	Cranial irradiation, ith. MTX	Prospective studies	[27]

## Data Availability

The data presented in this study are available on request from the corresponding author. The data are not publicly available due to ongoing analysis.

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
