# Peer review of "Cranial MRI in Childhood Acute Leukemia during Treatment and Follow-Up Including the Impact of Intrathecal MTX—A Single-Center Study and Review of the Literature"

_cancers, 2022, doi:10.3390/cancers14194688_

Round 1

Reviewer 1 Report

In a large single center cohort, Mergen and colleagues analyzed morphologic alterations of the brain and variations of the ventricular width (GDAH) in cranial MRIs of children with acute leukemias. They conducted their analyses without correlating their findings to neurocognitive deficits.

As follow-up MRI scans were only performed in patients with neurologic findings, half of the patients only received a baseline MRI at diagnosis. In the symptomatic half of patients, the authors could demonstrate that pathomorphological findings on MRIs were significantly higher in patients with ALL and younger patients (< 6 years). Of interest, females, irradiated or relapse patients showed only  tendencies towards higher prevalences of pathomorphological findings. GDAH increased mainly in boys and younger children (< 6 years), in those with ALL, high-risk leukemia, recurrence and if other pathomorphologies were detected in cranial MRIs. Of importance, the authors noted a potential for normalization of observed increases. The number of intrathecal MTX doses (≤ 12 vs > 12) showed no correlation to morphologic alterations besides a significant increase in GDAH between the first and last follow-up MRI in ALL patients receiving > 12 doses of intrathecal MTX.

The manuscript is elegantly written and was a pleasure to read: clear in its scientific flow, serious in its attempts to consolidate findings, logic in its linkage to already existing information on the issue, and relevant in its discussion of the findings. I have only one suggestion for improvement of this paper:

1. The BFM trials 2000 and 2009 on treatment of ALL included randomized questions on glucocorticoid use and extended use of PEG asparaginase, respectively. Did the authors had a chance to analyze a potential impact of the randomized regimens on cerebral pathomorphologies?

2. When defining the different periods of observation, please assure that it becomes clear how the baseline MRIs at diagnosis were dealt with. Have they (always) been included in P1?

3. Please re-check numbers in the manuscript for accuracy and consistency. For example, in the first line of the paragraph on WMC (line 165), 171 MRIs are mentioned. Please also check again for typos (e.g., line 88).

Author Response

We thank the reviewer for the time invested and for the important contributions to improve the manuscript. In the following we answer the comments one by one and explain our changes to the manuscript.

1.The BFM trials 2000 and 2009 on treatment of ALL included randomized questions on glucocorticoid use and extended use of PEG asparaginase, respectively. Did the authors had a chance to analyze a potential impact of the randomized regimens on cerebral pathomorphologies?

Response:

Thanks for this valuable question. We support its importance, but to answer it, a larger database is needed to find differences. Unfortunately, the number of our patients in BFM trial 2000 is only 24 and in trial 2009 only 42. In BFM 2000 the randomization between Dexamethasone and Prednisone was stopped in 2007, when we started to include patients in our cohort. In addition, all patients did receive Dexamethasone in protocol II. This makes it unlikely to find a difference. In BFM trial 2009 prolonged Asparaginase is only randomized in pre-B ALL during protocol II in the median risk group. This again reduces the number of patients being randomized. In summary these number are in both trials too low to find a significant difference. We added the following sentence in the discussion: “Unfortunately, the number of patients was too small to compare the influence of the type of corticosteroid (dexamethasone versus prednisone, ALL-BFM 2000) or the prolongation of asparaginase (AIEOP-BFM ALL 200) in randomized ALL trials patients.”(lines 803ff).

2. When defining the different periods of observation, please assure that it becomes clear how the baseline MRIs at diagnosis were dealt with. Have they (always) been included in P1?

Response:

The baseline MRIs are all included in P1. This is now mentioned in materials and methods: “All patients received their first MRI up to 30 days after diagnosis (induction period P1). Those without neurological symptoms (47 patients; 50%) at any time received only one cMRI during P1. In 29 patients (30.1%) two, in 7 (7.4%) three and in 11 (11.7%) four cMRIs were performed. Overall, we analyzed 170 cMRI scans (146 in ALL, 24 in AML).” (lines 74ff).

3. Please re-check numbers in the manuscript for accuracy and consistency. For example, in the first line of the paragraph on WMC (line 165), 171 MRIs are mentioned. Please also check again for typos (e.g., line 88).

Response:

Thanks for pointing to these mistakes. We have checked the whole manuscript again for typos, accuracy, and consistency. Changes are seen in track mode of the updated manuscript.

Reviewer 2 Report

In this study, Mergen and colleagues review their institution experience with MRI CNS abnormalities in 94 children ( 170 MRIs)with acute leukemia who received IT MTX. They find abnormalities in 50%. They detail the abnormalities in the paper. They find that the risk of abnormalities increases to 75% after XRT.

This is an important study which can be improved with 2 additional suggestions:1) an expert English editorial to simplify the sentences;2) reducing the details related to techniques and Tables. I particular, I have the following suggestions:

  1. Abstract – nice and concise. No suggestions
  2. Graphical abstract – is it needed or required? If not I would delete
  3. Introduction—Nice and concise
  4. Materials and Methods ; lines 91-103 – Needed? Can it move into Supplement?
  5. Figure 2—Move to Supplement
  6. Results – A bit difficult and verbose. This is where editorial help is needed.
  7. Table 1 – Too complicated to understand. Simplify or summarize in Text and move to Supplement
  8. Figures 3 and 4. Not important. Summarize under Results and move to Supplement
  9. Tables 2 and 3 and 4
  10. There are 2 Tables 4. I suggest the second Table 4 also be summarized in Text and move to Supplement.  – Again difficult to understand – Summarize in text and move to Supplement. Or, simplify the Tables
  11. Table 5 – OK . Can you make it easier to follow?
  12. Discussion – Fine. Maybe shorten by 1/3

Author Response

We thank the reviewer for the time invested and for the important contributions to improve the manuscript. In the following we answer the comments one by one and explain our changes to the manuscript.

This is an important study which can be improved with 2 additional suggestions:1) an expert English editorial to simplify the sentences; 2) reducing the details related to techniques and Tables. In particular, I have the following suggestions:

Response:

We have let a native speaker to simplify the sentences. Our changes to details related to techniques and tables and to other suggestions are explained below.

1.Abstract – nice and concise. No suggestions

Response:Thanks.

2.Graphical abstract – is it needed or required? If not, I would delete

Response:

The graphical abstract is asked by the journal. The editor can decide if needed or not. We, as authors, accept to delete it. On the other hand, we believe that it nicely summarizes important results of the paper.

3.Introduction—Nice and concise

Response: Thanks.

4.Materials and Methods; lines 91-103 – Needed? Can it move into Supplement?

Response:

Lines 93-101 are deleted in the text and added as a supplemental text (S1) named ‘Details of the cMRI scans’

5.Figure 2—Move to Supplement

Response:

Figure 2 is moved to the Supplement as Supplemental figure S1.

6.Results – A bit difficult and verbose. This is where editorial help is needed. 

Response:

Thanks for this comment. We have updated the results section. Changes can be found in the track mode of the updated manuscript.

7.Table 1 – Too complicated to understand. Simplify or summarize in Text and move to Supplement

Response:

We agree that table 1 contains a lot of information and some time is needed to understand all its details. We have restructured the table for better understanding and moved this table to the supplement as supplemental table S1.

Relevant information from this table is given in the results section.

8.Figures 3 and 4. Not important. Summarize under Results and move to Supplement

Response:

We are convinced that figure 3 and 4 (now figure 2 and 3) are important and therefore not moved to the supplement. They highlight ventricular width (GDAH) of all MRIs for different time periods (P1, P2, P3) and categories (fig. 2) and show the significant individual changes of GDAH between different time periods (fig. 3). This spares text in the result section. For better understanding figure 4 (now figure 3)  is renewed.

9.Tables 2 and 3 and 4

Response:

Tables 2, 3 and 4 are renamed to table 1,2 and 3. 

10.There are 2 Tables 4. I suggest the second Table 4 also be summarized in Text and move to Supplement.  – Again, difficult to understand – Summarize in text and move to Supplement. Or, simplify the tables

Response:

Thanks for highlighting this mistake. We have renumbered all the tables. As supplemental table S2 (former supplemental table S1) contains all details of table 4 and 5 we feel that table 4 should stay in the manuscript as it summarizes the former supplemental table S1 (now supplemental table S2).

11.Table 5 – OK. Can you make it easier to follow?

Response:

In our understanding there is no easier way to follow the table if it should contain all the information. We believe that the complete information is necessary.

12.Discussion – Fine. Maybe shorten by 1/3

Response:

We have tried to shorten as much as possible. See the changes in track mode of the updated manuscript.

Round 2

Reviewer 2 Report

Accept